# Burst Stimulation of the Thoracic Spinal Cord near a Cardiac Pacemaker in an Elderly Patient with Postherpetic Neuralgia: A Case Report

**DOI:** 10.3390/medicina57040337

**Published:** 2021-04-01

**Authors:** Yeon Joo Lee, Myoung Hoon Kong, Sang Sik Choi, Yong Deok Kwon, Mi Kyoung Lee, Chung Hun Lee

**Affiliations:** Department of Anesthesiology and Pain Medicine, Korea University Medical Center, Guro Hospital, Gurodong Road 148, Guro-Gu, Seoul 08308, Korea; pipipipig@naver.com (Y.J.L.); kong0331@korea.ac.kr (M.H.K.); clonidine@empal.com (S.S.C.); kwon129@naver.com (Y.D.K.); mknim@hotmail.com (M.K.L.)

**Keywords:** spinal cord stimulation, permanent cardiac pacemaker, post-herpetic neuralgia

## Abstract

New developments in spinal cord stimulation (SCS) have improved the treatment of patients with chronic pain. Although the overall safety of modern SCS has been established, there are no published reports regarding safety considerations when implanting a burst-mode spinal cord stimulator in patients with permanent cardiac pacemakers (PCPs). An 80-year-old man with a complete atrioventricular block implanted with a PCP was considered as a candidate for burst-mode SCS due to well-established postherpetic neuralgia (>180 days after rash). Cardiac monitoring during the burst-mode spinal cord stimulator trial and insertion did not indicate any interference. After the insertion of the burst-mode spinal cord stimulator, the patient showed functional improvement and significant pain relief. The safety of traditional tonic-mode SCS in patients with PCP has been previously reported. This is the first case report describing the safe and effective use of burst-mode SCS in a patient with PCP.

## 1. Introduction

Herpes zoster (HZ) is caused by the reactivation of latent varicella-zoster virus residing in the dorsal root ganglia or trigeminal ganglion after the first infection. This leads to abnormal nociceptor sensitization and hyperexcitability of the central nervous system. The frequency of reactivation increases in elderly patients and individuals in an immunosuppressed state [1]. The acute phase of HZ is defined as the period within 30 days after the emergence of the rash. When the pain persists for >30 days after the onset of the rash, it is defined as postherpetic neuralgia (PHN) [2,3], and if it persists for >180 days after the onset of the rash, it is considered a well-established PHN. The rate of successful management of well-established PHNs has been reported to be very low [2].

PHN can easily occur due to risk factors such as old age, despite treatment in the acute phase of HZ [4]. PHN is an intractable disease that is accompanied by severe neuropathic pain and a lower quality of life. It has become a more generalized disease, as 70% of patients >50 years of age complain of pain beyond one month after the disappearance of the skin rash, and 50% of patients >70 years of age experience pain for up to one year [2]. PHN treatment focuses on improving the quality of life by ensuring adequate pain relief [5]. This requires active treatment in a timely manner. Antiviral medication should be started as soon as possible after the clinical signs of HZ appear [5]. Steroids, tricyclic antidepressants, antiepileptics, and opioids are used for pain reduction [5]. Interventional treatments include epidural block, paravertebral block, sympathetic block, pulsed radiofrequency, and spinal cord stimulation (SCS) [5]. While the implantation of an SCS device is the most invasive procedure compared to other interventions, it may be considered when medical therapy or other interventional procedures have proven ineffective [6].

Previous case reports indicate that caution should be exercised when performing SCS system implantation in patients with permanent cardiac pacemakers (PCPs) [7]. A previous study reported PCP inhibition in 10 patients who received both SCS and unipolar PCP [8]. In the 2000s, safe methods of performing SCS in patients with PCP were reported, along with evidence of stability, including the use of the bipolar mode [7,9,10]. However, research on the safety of new modes of SCS, such as burst mode, in patients with PCP is still insufficient [11,12].

We report a case in which effective and safe pain control was achieved without complications using burst-mode SCS in an elderly patient with PCP and well-established PHN in the thoracic region.

## 2. Case Report

This study was approved by the Institutional Review Board of the Korea University Medical Center, Guro Hospital, Seoul, Republic of Korea (2019GR0050) on 11 February 2019. The patient was an 80-year-old man who had a PCP (Advisa DR MRI™ SureScan™ Model A2DR01, MEDTRONIC) implanted three months prior to admission because of complete atrioventricular block. The patient’s pacemaker was operated in a dual-chamber, rate-modulated pacing (DDDR) mode, and from a cardiac point of view, it was stable after insertion. The patient had been taking medication for hypertension for 10 years and suffered from chronic renal failure (stage 3). Two weeks prior to admission, he was diagnosed with acute HZ because he had a painful blister with a visual analog scale (VAS) score ≥9 on the second left thoracic dermatome, and was prescribed famciclovir and pain relievers, including pregabalin, analgesics (ibuprofen, codeine, acetaminophen), amitriptyline, and a lidocaine patch, but was admitted to the hospital because of persistent throbbing pain in the upper left arm, shoulder, and chest. After admission, we tried several treatment methods, such as epidural nerve block, continuous epidural catheter insertions, and C-arm-guided pulsed radiofrequency to control the zoster-related pain. However, none of these interventions was successful. The pain waxed and waned for a while, and finally, the excruciating pain (VAS 8) and sleep disturbance relapsed.

On day 180 of the rash, SCS was suggested to relieve the uncontrolled pain. All common risks of SCS (e.g., infection, hardware complications, and bleeding) were discussed with the patient and the patient’s caregiver in the course of obtaining informed consent. Additionally, we discussed the potential risks to PCP from SCS and their implications. We were unaware of the statistical likelihood of this risk. However, based on studies [7,9,10] that state that tonic mode SCS can function safely with PCP, studies [13] that state that burst mode SCS consumes less energy than tonic-mode SCS and theories that the difference in Hz is not much larger than that of tonic mode SCS, but rather reduces the frequency of stimulation, we believed that the risk was low. After being fully informed of the risks and benefits of the procedure, the patient and his caregiver agreed to the SCS system insertion.

The SCS system (Implantable pulse generator (IPG): prodigy MRI, IPG, ST. JUDE Medical, Abbott, Chicago, IL, USA) trial was conducted in the operating room under monitored anesthesia management. While the patient was in a prone position, percutaneous local anesthesia was administered using 1% lidocaine using an aseptic technique. Using the loss-of-resistance method, a 16-gauge Tuohy needle was positioned between the T10 and T11 vertebrae, guided by a C-arm. The target area was confirmed by inserting a guidewire. Subsequently, the electrodes (percutaneous lead 8 pole, ST. JUDE Medical, Minneapolis, MN, USA) were positioned from T1/2 to T3/4 and directed toward the head (Figure 1a–c).

Pacemaker sensitivity and possible interference between PCP and SCS were tested during the SCS trial in the operating room. This test was conducted in collaboration with a cardiologist using a pacemaker monitoring device that allowed for full monitoring of the pacemaker. The main principle is to maximize the likelihood of interference temporarily. PCP ventricular detection was set to the bipolar mode, and the ventricular detection threshold was lowered to the lowest acceptable level. In addition, we tried both tonic-mode SCS and burst-mode SCS during the SCS trial. The tonic mode SCS parameters consisted of 50 Hz, pulse width 500 (maximum value). The burst-mode SCS parameters consisted of a 40 Hz burst rate, 25 ms duration, 1 ms pulse width, and 5-pulse burst train, standardized to an intra-burst rate of 500 Hz. The SCS output was in the bipolar mode, and the maximum permissible stimulation energy (approximately 120% of the clinically used stimulation level) output was reached.

No interference between the PCP and SCS was observed during the SCS trial in the operating and recovery rooms (Figure 2a–c).

Afterwards, EKG monitoring in the ward was continuously performed using lead 2 and continued for 2 days after the SCS trial. During the two days, the patient did not complain of symptoms such as chest discomfort in tonic mode SCS and burst mode SCS, and no abnormal findings were observed on lead 2 EKG monitoring.

During SCS trials, weak paresthesia was observed on the left T2/3 skin segment by stimulation through the percutaneous nerve stimulator, and the previous throbbing pain and soreness in the upper arm, shoulder, front thorax, and left posterior T2 dermatome reduced from a VAS score of 8 to 3, especially in burst-mode SCS. During the 1 week period after SCS trials, the patient’s pain in burst mode SCS was reduced by a VAS score of 3–4. The patients reported pain reduction more significantly in burst mode SCS than in tonic mode SCS and opted for permanent implantation of the burst mode SCS system. Permanent implantation of the SCS system was performed one week after the SCS trial.

One month after insertion of the SCS system, the VAS score was consistent at 3, and the patient reduced the dose of narcotic analgesic (oxycodone 5 mg) three times a day to twice a day and did not complain of any particular discomfort at follow-up after one month. Three months after insertion of the SCS system, the VAS score increased to about 4, but the patient did not complain of a major obstacle in daily life. At follow-up 6 months after SCS system insertion, the patient showed pain relief of approximately 50% (VAS 4), and among his oral medications, amitryptyline was discontinued and the pregabalin dose was reduced from 75 mg, twice a day, to 50 mg, twice a day.

## 3. Discussion

We present a case in which an SCS system and PCP were simultaneously operated safely and effectively, even in burst-mode SCS, by inserting an SCS system with an 8-contact electrode (percutaneous lead 8 pole, ST. JUDE Medical) in the T2 dermatome in an elderly patient with a well-established PHN.

PHNs are more common in elderly patients [2,4]. SCS is a last-resort treatment option for patients with PHN when pain is no longer tolerable despite medication and interventional therapies [6]. Activation of large-diameter afferent nerve fibers due to electrical stimulation of the spinal cord can suppress small fierce pain input. This is known as the “Gate control theory of pain” [11]. Therefore, the use of SCS in PHNs has gradually expanded [11]. Furthermore, because elderly patients are more likely to develop heart disease, they are more likely to have implantable cardiac electronics than other patient populations [14].

The patient described here was an 80-year-old man with well-established PHN and PCP operating in the DDDR mode. We were unable to confirm the safety of inserting a burst-mode SCS system into the thoracic spine in patients >80 years of age using DDDR-mode pacemakers.

According to previous case reports, caution should be exercised when using SCS simultaneously with PCP because of the risk of inappropriate inhibition (i.e., the SCS signal may be misinterpreted by the PCP as a normal R wave) [8]. A study of 10 patients with both SCS and PCP reported that PCP inhibition was found, and that the overall safety of the combined devices could not be guaranteed. It was reported that patients who showed inhibition in this study used both devices in the unipolar mode [8]. Based on this, there is concern that SCS may interfere with the functioning of these implantable cardiac pacemakers [8,15,16]. Therefore, it has been proposed that the following points must be considered for the safe use of PCP and traditional tonic SCS simultaneously [7,8,9,10,15,16,17,18].

First, the pacemaker must be programmable in terms of sensitivity, and it must be set to the lowest sensitivity (considering the R-wave amplitude) to accurately detect electrical signals from the heart. Second, a bipolar pacemaker system is preferable because it is less sensitive to external electrical signals. Third, the nerve stimulation system should preferably be bipolar. (The electrode spacing should be as small as possible). Fourth, pacing or sensing leads and neuro leads should be spatially separated as much as possible. Fifth, before treatment, the operator should know the pacing mode and interference frequency of the pacemaker. Sixth, during the neural system implant procedure, a test should be performed.

Recent developments in SCS systems include the addition of various modes, such as the burst mode. [11]. The conventional SCS paradigm uses a tonic stimulation pattern at low frequencies (typically 30–70 Hz) to elicit a comfortable sensation in the area of pain [19]. The concept of burst-mode SCS, which was introduced in 2010 by DeRidder et al. [20], consists of a burst of five impulses spaced 1 ms at a frequency of 500 ms and applied at 40 Hz. The concept for the burst-mode SCS came from the original observation of the talamo-cortical firing pattern with the ability to strengthen synaptic connections [21,22]. In other words, studies examining the impact of burst stimulation on a class of cortical neurons have shown that thalamic bursts strongly activate cortical circuits [22]. They argued that the interval between each burst is important for creating an improved cortical response [22]. This burst-mode SCS pattern has been shown to result in statistically superior pain relief compared to tonic stimulation in a large prospective randomized controlled trial [12]. In addition, it has been reported that burst-mode SCS does not require stimulation-induced paresthesia to relieve pain [20]. However, it is not yet known whether burst-mode SCS works safely with a PCP without mutual interference.

This burst-mode SCS trial was conducted based on the patient’s strong request and the assumption of previous study results that PCP and SCS would not interfere with each other. However, this was based on general reasoning rather than clear evidence. Fortunately, the pacemaker did not show any abnormal functioning, and the patient’s pain was properly controlled during the 1-year follow-up period after burst-mode spinal cord stimulator implantation. However, additional studies with longer observation periods and larger sample sizes are needed to determine effective pain relief and safe simultaneous maintenance of burst-mode SCS and PCP.

## 4. Conclusions

This case report indicates that burst mode SCS can be an effective and safe treatment for patients with intractable PHN with PCPs. Thorough monitoring to detect the interference between the two devices is a prerequisite for the simultaneous and safe use of burst-mode SCS and PCP.

## Figures and Tables

**Figure 1 medicina-57-00337-f001:**
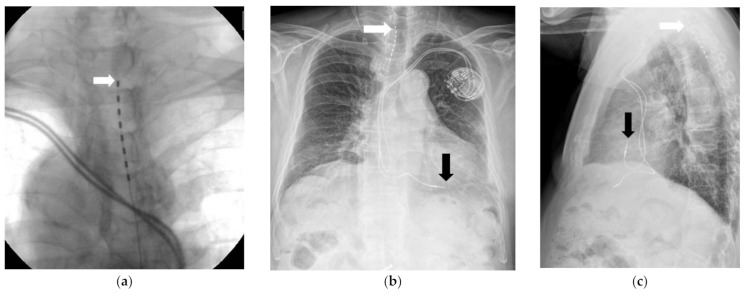
(**a**) Spinal cord stimulation trial 180 days after the onset of rash. The arrow indicates the lead of the spinal cord stimulator. (**b**) A chest postero-anterior view, obtained on the next day, after the spinal cord stimulation trial. The white arrow indicates the lead of the spinal cord stimulator. The black arrow indicates the lead of the permanent cardiac pacemaker. (**c**) The chest lateral view obtained on the next day after the spinal cord stimulation trial. The shortest distance between the lead of the spinal cord stimulator and the lead of the cardiac pacemaker was 163.41 mm. The white arrow indicates the lead of the spinal cord stimulator. The black arrow indicates the lead of the permanent cardiac pacemaker.

**Figure 2 medicina-57-00337-f002:**
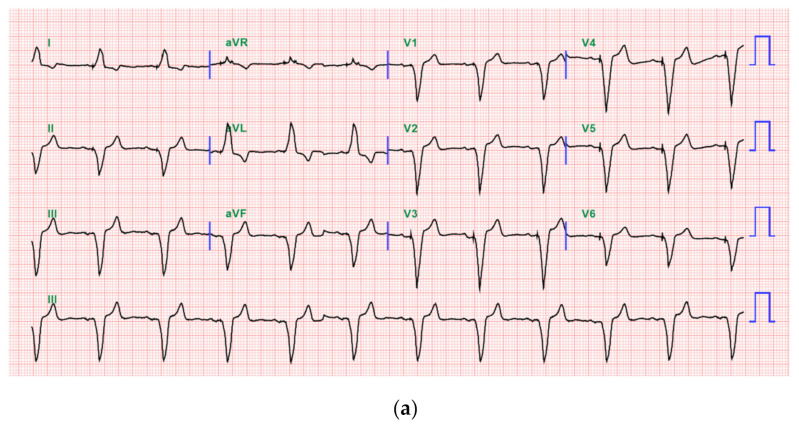
(**a**) 12 lead electrocardiogram (EKG) picture before spinal cord stimulation (SCS) trial. (**b**) 12 lead EKG picture in burst mode performed in recovery room after SCS trial. (**c**) 12 lead EKG picture in tonic mode performed in recovery room after SCS trial.

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
