# Peer review of "Burst Stimulation of the Thoracic Spinal Cord near a Cardiac Pacemaker in an Elderly Patient with Postherpetic Neuralgia: A Case Report"

_medicina, 2021, doi:10.3390/medicina57040337_

Round 1

Reviewer 1 Report

This is an interesting case report and I read with great interest. However, there are several points that require further clarification in order for the readers to appreciate the procedures used in this case.

    1. The authors stated in the Introduction that they employed a new model of spinal cord stimulation (SCS) but did not explain what is in the new model. Although the authors provided the stimulation parameter in the discussion, it would be much appreciated for this information to be described earlier in the manuscript. Particularly there was a study reporting safety concerns around SCS combined with a permanent cardio pacemaker. Did the authors have evidence to believe that the new model would be safer? Please comment on this point and include clarification in the manuscript.
    2. The patient had received several types of pain management interventions before the SCS was given. It is possible that the previous pain management interventions may interact with the SCS procedure. Please provide detail in the dosage and duration of other pain management intervention. Were other interventions given together with the SCS? How did the patient gradually reduce the amount of pain killers used along the course of time after the SCS procedure?
   3. The authors included a previous study reporting safety concerns around SCS combined with permanent cardio pacemakers. However, later on in the manuscript the authors stated that there is insufficient evidence to evaluate the risk and the cardiology team suggested the risk of SCS combined with the pacemaker was low. This seems to contradict to the current evidence. Please clarify on this point.
   4. How was the SCS stimulation parameter determined? Was the parameter being tested in the theatre while the patient was under local anaesthesia and awake? The authors reported a reduction in pain of the patient subjective visual analog scale during SCS. Please provide information on when the assessment was performed.
   5.  The authors described that an ECG monitor was used after the SCS was implanted and that there was no interference between the SCS and the pacemaker. It would be helpful to provide a figure showing the ECG reading. I assume that there might be SCS artefacts shown on the ECG monitor. Providing such figure will be valuable for other medical teams who wish to apply this method.
   6. The ECG monitor was on for two days. Was it monitoring 24 hours a day? Was there any unusual signals being detected even though the patient did not feel unwell or show symptoms? Please report detail in assessment results related to the safety of the procedure.
   7. The authors claimed that the effect of SCS on pain reduction was through the principle of the gate control theory of pain. Another pain management intervention shared the same principle is non-invasive TENS. Was TENS used before the decision for implanting SCS was made? Please comment on this point.
  8.  Page 5, line 165-167. Please elaborate the reasoning of point number 4 and 5. Were they the procedures related to a PCP implementation or SCS? Please clarify in the manuscript.
  9. Page 6, line 176. The authors claimed that the effect of SCS was through strengthening synaptic connections. This is different from the gate control theory of pain proposed earlier in the discussion. Was there evidence to support this claim?
   10. Page 6, line 182-184. I am unsure what the authors meant by regulating brain regions involving physical and emotional pain control. What exactly are the physical and emotional pain control the authors referred to? And what is the rationale behind that only burst SCS could achieve this effect? The SCS is a spinal stimulation and does not directly modulate the brain activation. Please revise this paragraph to avoid confusion.

Author Response

MEDICINA-1138842

Burst stimulation of the thoracic spinal cord near a cardiac pacemaker in an elderly patient with postherpetic neuralgia: A case report

Medicina

Response to reviewers’ comments

We appreciate the reviewer’s thoughtful comments and suggestions. We have made changes to the manuscript based on these recommendations and believe that these changes have improved the quality of our manuscript. Our point-by-point responses and descriptions of these changes are listed below.

We thank the reviewers and editor for their time.

Sincerely,

Chung Hun Lee

Department of Anesthesiology and Pain medicine,

Korea University Medical Center, Guro Hospital,

Gurodong Road 148, Guro-Gu, Seoul 08308, Republic of Korea

Review Comments to the Author

Reviewer #1

This is an interesting case report and I read with great interest. However, there are several points that require further clarification in order for the readers to appreciate the procedures used in this case.

1. The authors stated in the Introduction that they employed a new model of spinal cord stimulation (SCS) but did not explain what is in the new model. Although the authors provided the stimulation parameter in the discussion, it would be much appreciated for this information to be described earlier in the manuscript. Particularly there was a study reporting safety concerns around SCS combined with a permanent cardio pacemaker. Did the authors have evidence to believe that the new model would be safer? Please comment on this point and include clarification in the manuscript.

- Thank you for pointing this out. In the introduction (page 2, line 50), a new model of spinal cord stimulation was added to the “burst mode”. In addition, information on the stimulation parameters was transferred to the previous section and described.

Unfortunately, the authors had no evidence that the new model (burst mode SCS) would be safer when operated with PCP compared to the existing tonic mode SCS. However, compared to the previous tonic mode SCS, burst mode SCS has been reported to be more helpful in pain relief and less paresthesia [1]. In addition, in this case, when tonic and burst modes were used during the SCS trial, pain relief was greater in burst mode, so the patient preferred burst mode. Therefore, the patient used burst mode SCS and, fortunately, showed that both worked well without significant interference with the PCP. This has been added to the text and has been described.

2. The patient had received several types of pain management interventions before the SCS was given. It is possible that the previous pain management interventions may interact with the SCS procedure. Please provide detail in the dosage and duration of other pain management intervention. Were other interventions given together with the SCS? How did the patient gradually reduce the amount of pain killers used along the course of time after the SCS procedure?

- There were no other types of interventions provided with SCS, and only medication (oxycodone, amitryptyline, pregabalin) was administered concurrently with SCS. In addition, the amount of painkillers used over time after the SCS procedure in the patient is described in more detail in the text.

  • One month after insertion of the SCS system, the VAS score was consistent at 3, and the patient reduced the dose of narcotic analgesic (oxycodone 5 mg) from thrice a day to twice a day, and did not complain of any particular discomfort at follow-up after one month. Three months after insertion of the SCS system, the VAS score increased to about 4, but the patient did not complain of a major obstacle in daily life. At follow-up 6 months after SCS system insertion, the patient showed pain relief of approximately 50% (VAS 4), and among his oral medications, amitryptyline was discontinued and the pregabalin dose was reduced from 75 m, twice a day to 50 mg twice a day.

3. The authors included a previous study reporting safety concerns around SCS combined with permanent cardio pacemakers. However, later on in the manuscript the authors stated that there is insufficient evidence to evaluate the risk and the cardiology team suggested the risk of SCS combined with the pacemaker was low. This seems to contradict to the current evidence. Please clarify on this point.

- We found no evidence in the literature to assess the risk between burst mode SCS and PCP. However, the study confirmed that the risk between tonic mode SCS and PCP can be reduced through several methods [2-4]. Based on the fact that burst mode SCS has little difference in Hz than tonic mode SCS and rather reduces stimulation frequency, we thought that the possibility that burst mode SCS could interfere with PCP is not high. In addition, the energy required to deliver burst mode SCS has been reported to be less than that of tonic mode SCS in many patients, which is one of the reasons we judged less interference with PCPs [5]. However, this was due to general reasoning, not based on clear evidence, as the reviewer said, which may act as a limitation of this case report. Therefore, the above information has been revised and described in the case report section, and is additionally described as a limitation in the discussion section.

4. How was the SCS stimulation parameter determined? Was the parameter being tested in the theatre while the patient was under local anaesthesia and awake? The authors reported a reduction in pain of the patient subjective visual analog scale during SCS. Please provide information on when the assessment was performed.

- The SCS stimulation parameters were based on the reported reference for burst mode SCS, and the commonly used burst mode SCS stimulation parameters were used [5]. While the patient was awake under local anesthesia, the parameters and stimulation location were adjusted by talking with the patient. During the one week period after the SCS trial, we found the stimulation location and SCS mode where the patient's pain was most suppressed. In addition, the patient's subjective visual analog scale was evaluated daily during the one week period after the SCS trial. Based on this, it was judged that the pain reduction was greater in burst mode SCS than in tonic SCS, so a burst mode SCS permanent procedure was performed one week after the SCS trial. This has been added to the case report section. Thank you.

5. The authors described that an ECG monitor was used after the SCS was implanted and that there was no interference between the SCS and the pacemaker. It would be helpful to provide a figure showing the ECG reading. I assume that there might be SCS artefacts shown on the ECG monitor. Providing such figure will be valuable for other medical teams who wish to apply this method.

- Thank you for pointing this out. EKG (12 lead EKG) taken before the SCS trial, and EKG (12 lead EKG) in burst mode and tonic mode SCS in the recovery room were added to the text as shown in figures 2A, 2B, and 2C. In fact, we judged that before the SCS trial, the 12 lead EKG pictures in burst mode and tonic mode did not show much difference, and in both burst mode and tonic mode, the patient did not complain of symptoms such as chest discomfort.

6. The ECG monitor was on for two days. Was it monitoring 24 hours a day? Was there any unusual signals being detected even though the patient did not feel unwell or show symptoms? Please report detail in assessment results related to the safety of the procedure.

- The 12 lead EKG evaluation was conducted before and immediately after the SCS trial. Afterwards, EKG monitoring in the ward was continuously monitored for 2 days after the SCS trial as an EKG monitor that only saw lead 2. Fortunately, the patient did not complain of any symptoms such as chest discomfort for 2 days, and no abnormal findings were observed on lead 2 EKG monitoring. This has been added to the case report section. Thank you.

7. The authors claimed that the effect of SCS on pain reduction was through the principle of the gate control theory of pain. Another pain management intervention shared the same principle is non-invasive TENS. Was TENS used before the decision for implanting SCS was made? Please comment on this point.

- As the reviewer stated, non-invasive TENS can also be described as a gate control theory. However, we did not have much experience using TENS in patients with PHN. In addition, the patient continued to complain of severe pain despite invasive interventions such as continuous epidural block and RF for approximately 6 months. Therefore, the patient wanted a powerful and reliable pain relief method. Therefore, we performed SCS, of which we have experienced, and which is considered as an ultimate treatment for patients with post-herpetic neuralgia [6]. However, we think it is possible to consider implementing non-invasive TENS before SCS, as the reviewer stated. Thank you.

8. Page 5, line 165-167. Please elaborate the reasoning of point number 4 and 5. Were they the procedures related to a PCP implementation or SCS? Please clarify in the manuscript.

- Thank you for pointing this out. The text has been revised as follows to clarify points 4 and 5 on page 6, lines 179-181.

  • “Third, the nerve stimulation system should preferably be bipolar. (The electrode spacing should be as small as possible). Fourth, the pacing or sensing leads of PCP and neuro leads of SCS should be spatially separated as much as possible.”

9. Page 6, line 176. The authors claimed that the effect of SCS was through strengthening synaptic connections. This is different from the gate control theory of pain proposed earlier in the discussion. Was there evidence to support this claim?

- The therapeutic concept of spinal cord stimulation (SCS) for refractory neuropathic pain is based on gate-control theory, which theorizes that stimulation of large A-beta fibers could close the gateway to pain signaling [5]. The mechanism of SCS has also been reported to include a combination of spinal and supraspinal structures [7,8]. The ascending dorsal column fibers in the spinal cord and the descending supraspinal projections are reported to use an opioidergic and serotonergic modulatory system to suppress pain [9-11]. The involvement of large A-beta fibers during tonic mode SCS trials is thought to be expressed as paresthesia; therefore, the best predictor of effective SCS treatment is paresthesia coverage over the painful areas [12,13].

However, it has been reported that the mechanism of action of burst mode SCS is different from that of tonic mode SCS. Both tonic mode SCS and burst mode SCS cause activation of ascending fibers that form synapses in the thalamus and continue to protrude laterally into the sensory cortex.

The difference between burst mode SCS and tonic mode SCS is that another post-thalamic pathway is activated, which medially projects into the cingulate gyrus and other limbic system structures [14,15].

In particular, one study investigated bursting thalamic impulses to their terminus at the thalamocortical synapse of the awake rabbit, and examined their influence on a class of somatosensory cortical neurons [16]. In that study, it was shown that thalamic bursts strongly activates cortical circuits. In addition, it was reported that the initial impulse of each burst significantly improved the ability to induce a cortical action potential, and the later impulses in the burst further increase the probability of eliciting spikes. In other words, they argued that the interval between each burst is important in generating an improved cortical response [16]. Based on this study, the authors have described the phrase “The concept for the burst-mode SCS came from the original observation of the talamo-cortical firing pattern with the ability to strengthen synaptic connections.” in page 6, line 191. However, as the reviewer said, it was judged that this text alone was insufficient; therefore, a brief description of burst mode SCS was added to page 6, line 192-195.

10. Page 6, line 182-184. I am unsure what the authors meant by regulating brain regions involving physical and emotional pain control. What exactly are the physical and emotional pain control the authors referred to? And what is the rationale behind that only burst SCS could achieve this effect? The SCS is a spinal stimulation and does not directly modulate the brain activation. Please revise this paragraph to avoid confusion.

- A study by Deer et al. showed that localized electroencephalography showed that burst mode SCS activates both the lateral and medial cerebral projecting pathways [5]. The lateral pathway activates the sensory cortex, which interprets the discriminatory components of pain, such as location, type, and intensity. The medial pathway projects to the dorsal anterior cingulate cortex, which determines pain awareness, vigilance, and emotional responses to pain [14,17].

It has been reported that burst-driven limbic activation by this medial pathway plays a role in treating the affective component of pain as well as overall pain perception [18]. However, as stated by the reviewer, the above content has been deleted from the main text as it is judged that it may be confusing as the content reported only in some studies. Thank you.

References

  1. Deer, T.; Slavin, K.V.; Amirdelfan, K.; et al. Success Using Neuromodulation with BURST (SUNBURST) Study: Results from a Prospective, Randomized Controlled Trial Using a Novel Burst Waveform. Neuromodulation 2018, 21, 56-66.
  2. Kosharskyy, B.; Rozen, D. Feasibility of Spinal Cord Stimulation in a Patient with a Cardiac Pacemaker. Pain Physician 2006, 9, 249-251.
  3. Hoelzer, B.; Burgher, A.; Huntoon, M. Thoracic Spinal Cord Stimulation for Post-Ablation Cardiac Pain in a Patient with Permanent Pacemaker. Pain Pract 2008, 8, 110-113.
  4. Ekre O.; Börjesson M.; Edvardsson N.; Eliasson T.; Mannheimer, C. Feasibility of Spinal Cord Stimulation in Angina Pectoris in Patients with Chronic Pacemaker Treatment for Cardiac Arrhythmias. Pacing Clin Electrophysiol 2003, 26, 2134-2141.
  5. Deer, T.R.; Campos, L.W.; Pope, J.E. Evaluation of Abbott’s BurstDR Stimulation Device for the Treatment of Chronic Pain. Expert Rev Med Devices 2017, 14, 417–422.
  6. Lin, C.S.; Lin, Y.C.; Lao, H.C.; Chen, C.C. Interventional Treatments for Postherpetic Neuralgia: A Systematic Review. Pain Physician 2019, 22, 209-228.
  7. Saadé, N.E.; Jabbur, S.J. Nociceptive behavior in animal models for peripheral neuropathy: spinal and supraspinal mechanisms. Prog Neurobiol 2008, 86, 22–47.
  8. Barchini, J.; Tchachaghian, S.; Shamaa, F.; et al. Spinal segmental and supraspinal mechanisms underlying the pain-relieving effects of spinal cord stimulation: an experimental study in a rat model of neuropathy. Neuroscience 2012, 215, 196–208.
  9. Sikandar, S.; Bannister, K.; Dickenson, A.H. Brainstem facilitations and descending serotonergic controls contribute to visceral nociception but not pregabalin analgesia in rats. Neurosci Lett 2012, 519, 31–36.
  10. Fields, H. State-dependent opioid control of pain. Nat Rev Neurosci 2004, 5, 565–575.
  11. Song, Z.; Ultenius, C.; Meyerson, B.A.; et al. Pain relief by spinal cord stimulation involves serotonergic mechanisms: an experimental study in a rat model of mononeuropathy. Pain 2009, 147, 241–248.
  12. Ochoa, J.L.; Torebjörk, H.E. Paraesthesiae from ectopic impulse generation in human sensory nerves. Brain 1980, 103, 835–853.
  13. North, R.B.; Ewend, M.G.; Lawton, M.T.; et al. Spinal cord stimulation for chronic, intractable pain: superiority of “multi-channel” devices. Pain 1991, 44, 119–130.
  14. de Ridder, D.; Vancamp, T.; Lenders, M.W.P.M.; et al. Is preoperative pain duration important in spinal cord stimulation? A comparison between tonic and burst stimulation. Neuromodulation 2015, 18, 13–17.
  15. de Ridder, D.; Plazier, M.; Kamerling, N.; et al. Burst spinal cord stimulation for limb and back pain. World Neurosurg 2013, 80, 642–649.
  16. Swadlow, H.A.; Gusev, A.G. The Impact of ‘Bursting’ Thalamic Impulses at a Neocortical Synapse. Nat Neurosci 2001, 4, 402–408.
  17. de Ridder, D.; Vanneste, S. Response: a systematic evaluation of burst spinal cord stimulation for chronic back and limb pain. Neuromodulation 2016, 19, 785–786.
  18. Friedman, A.; Friedman, Y.; Dremencov, E.; et al. VTA dopamine neuron bursting is altered in an animal model of depression and corrected by desipramine. J Mol Neurosci 2008, 34, 201–209.

Reviewer 2 Report

The authors present an interesting manuscript concerning the simultaneous use of Spinal Cord Stimulation (SCS) and a permanent cardiac pacemaker (PCP). The authors conclude that effective SCS stimulation using a non-conventional modern stimulation paradigm, namely Burst stimulation, did not interfere with PCP function. I find the article of value for readers working within this field. There are, however, a few points that I wish to bring up:

  • I guess that on line 30 the authors have missed an “if” between “and” and “it persists for …”.
  • I consider it of value for the readers to be informed of product name and manufacturer of the implanted PCP as well.
  • Famciclovir is an antiviral substance and should not be part of a list of “pain relievers”.
  • I see no reason for the amount of detail used to present the treatment methods utilized before SCS. Figure 1 and 2 do not add to the valuable information and could be omitted.
  • The time course of the SCS trial is not clear to me. On line 116-117 the authors write: “Pacemaker sensitivity and possible interference between PCP and SCS were tested before surgery”, but is this the case? The description on lines 92-99 rather hints to that SCS was implanted first. In lines 128-129 the authors write: “Permanent implantation of the SCS system was per-128 formed one week after the SCS trial.”. This description is difficult to follow. If the authors started out with a trial lead and later performed a permanent implantation, then they should specify this (if so, did they use a trial lead or a permanent lead? For how long did they test? They state that permanent implantation was performed one week after the trial but was this one week after removal of a percutaneous trial lead or one week after implantation of a permanent lead – or something else?). If they did some other way, please specify that. Was the testing in the sitting position referred to on line 125 done as part of the implantation procedure or performed separately?
  • The Discussion section should be substantially shortened. This is a case report and I fail to see why it should contain lengthy descriptions on the background for SCS in general or the rationale and results of Burst stimulation mode.

Author Response

MEDICINA-1138842

Burst stimulation of the thoracic spinal cord near a cardiac pacemaker in an elderly patient with postherpetic neuralgia: A case report

Medicina

Response to reviewers’ comments

We appreciate the reviewer’s thoughtful comments and suggestions. We have made changes to the manuscript based on these recommendations and believe that these changes have improved the quality of our manuscript. Our point-by-point responses and descriptions of these changes are listed below.

We thank the reviewers and editor for their time.

Sincerely,

Chung Hun Lee

Department of Anesthesiology and Pain medicine,

Korea University Medical Center, Guro Hospital,

Gurodong Road 148, Guro-Gu, Seoul 08308, Republic of Korea

Review Comments to the Author

Reviewer #2

The authors present an interesting manuscript concerning the simultaneous use of Spinal Cord Stimulation (SCS) and a permanent cardiac pacemaker (PCP). The authors conclude that effective SCS stimulation using a non-conventional modern stimulation paradigm, namely Burst stimulation, did not interfere with PCP function. I find the article of value for readers working within this field. There are, however, a few points that I wish to bring up. :

I guess that on line 30 the authors have missed an “if” between “and” and “it persists for …”.

- Thank you for pointing this out. In Page 1, line 30, we have added “if” between “and” and “it persists for….”

I consider it of value for the readers to be informed of product name and manufacturer of the implanted PCP as well.

- Thank you for pointing this out. On Page 2, line 58, the product name and manufacturer of the PCP to which the patient was implanted (Advisa DR MRI™ SureScan™ Model A2DR01, MEDTRONIC) was additionally described.

Famciclovir is an antiviral substance and should not be part of a list of “pain relievers”.

- Thank you for pointing this out. As stated by the reviewer, “Famciclovir” and “pain relievers” were listed separately on Page 2, line 65.

I see no reason for the amount of detail used to present the treatment methods utilized before SCS. Figure 1 and 2 do not add to the valuable information and could be omitted.

- As the reviewer said, Figures 1 and 2 are not data showing valuable information in this case report, so they have been deleted from the text. Thank you.

The time course of the SCS trial is not clear to me. On line 116-117 the authors write: “Pacemaker sensitivity and possible interference between PCP and SCS were tested before surgery”, but is this the case? The description on lines 92-99 rather hints to that SCS was implanted first. In lines 128-129 the authors write: “Permanent implantation of the SCS system was per-128 formed one week after the SCS trial.”. This description is difficult to follow. If the authors started out with a trial lead and later performed a permanent implantation, then they should specify this (if so, did they use a trial lead or a permanent lead? For how long did they test? They state that permanent implantation was performed one week after the trial but was this one week after removal of a percutaneous trial lead or one week after implantation of a permanent lead – or something else?). If they did some other way, please specify that. Was the testing in the sitting position referred to on line 125 done as part of the implantation procedure or performed separately?

- Thank you for pointing this out. During the SCS trial in the operating room, with the presence of a cardiologist, possible interference between the SCS and PCP was observed with EKG monitoring. When stimulating several positions of the SCS lead in the operating room and performing the tonic and burst modes, the patient did not complain of special chest discomfort symptoms, and the patient's electrocardiogram (lead 2) did not show any special abnormalities. Subsequently, 12 lead EKG was performed with tonic mode SCS and burst mode SCS in the recovery room, and there was no significant change compared to before the SCS trial. After the patient moved to the ward, tonic mode SCS and burst mode SCS were applied alternately to the patient, and a close observation was made for 48 hours with lead 2 EKG, but no abnormality was observed.

One week after the SCS trial, the patient underwent permanent SCS implantation. Based on this, the authors judged “surgery” as a permanent SCS procedure and described it as “Pacemaker sensitivity and possible interference between PCP and SCS were tested before surgery.” on page 4, line 117 of the main text. However, as stated by the reviewer, referring to this as a preoperative test could be confusing, so on page 3, line 106, the contents of the test were revised and described in a clearer manner. In addition, the phrase “Pacemaker sensitivity and possible interference between PCP and SCS were tested before surgery.” was revised and described as “Pacemaker sensitivity and possible interference between PCP and SCS were tested during SCS trial in operating room.” In addition, on page 3-5, lines 106-132, the above-mentioned contents have been revised and described clearly.

The Discussion section should be substantially shortened. This is a case report and I fail to see why it should contain lengthy descriptions on the background for SCS in general or the rationale and results of Burst stimulation mode.

- Thank you for pointing this out. However, this case report is a report on the possibility of mutual interference between PCP and burst mode SCS, and the authors thought that it is necessary to explain the application of burst mode SCS to patients rather than tonic mode SCS. In addition, it is judged that burst mode is not widely used yet, so the rationale and results of burst stimulation mode are summarized in the discussion section. However, as the reviewer said, additional theories and results for burst stimulation that are not largely related to this case have been deleted from the discussion of the text. Thank you.

  • Deleted part: In addition to affecting pain intensity, burst-mode SCS has been reported to have a positive effect on important aspects of chronic pain conditions such as pain vigilance, pain catastrophes, and depression [18,22,23]. It is explained that burst SCS works by altering the physical and emotional components of pain, perhaps by regulating brain regions involved in physical and emotional pain processing [24,25].
